# Multimodality Helps Few-shot 3D Point Cloud Semantic Segmentation

**Zhaochong An**[1], **Guolei Sun**[2*], **Yun Liu**[3*] **Runjia Li**[4], **Min Wu**[5], **Ming-Ming Cheng**[3],
**Ender Konukoglu**[2], **and Serge Belongie**[1]

[1] Department of Computer Science, University of Copenhagen
[2] Computer Vision Laboratory, ETH Zurich
[3] College of Computer Science, Nankai University
[4] Department of Engineering Science, University of Oxford
[5] Institute for Infocomm Research, A*STAR

## Abstract

Few-shot 3D point cloud segmentation (FS-PCS) aims at generalizing models to segment novel categories with minimal annotated support samples. While existing FS-PCS methods have shown promise, they primarily focus on unimodal point cloud inputs, overlooking the potential benefits of leveraging multimodal information. In this paper, we address this gap by introducing a multimodal FS-PCS setup, utilizing textual labels and the potentially available 2D image modality. Under this easy-to-achieve setup, we present the **M**ulti**M**odal **F**ew-**S**hot **S**egNet (**MM-FSS**), a model effectively harnessing complementary information from multiple modalities. MM-FSS employs a shared backbone with two heads to extract intermodal and unimodal visual features, and a pretrained text encoder to generate text embeddings. To fully exploit the multimodal information, we propose a **M**ultimodal **C**orrelation **F**usion (**MCF**) module to generate multimodal correlations, and a **M**ultimodal **S**emantic **F**usion (**MSF**) module to refine the correlations using text-aware semantic guidance. Additionally, we propose a simple yet effective **T**est-time **A**daptive **C**ross-modal **C**alibration (**TACC**) technique to mitigate training bias, further improving generalization. Experimental results on S3DIS and ScanNet datasets demonstrate significant performance improvements achieved by our method. The efficacy of our approach indicates the benefits of leveraging commonly-ignored free modalities for FS-PCS, providing valuable insights for future research. The code is available at this link.

## 1 Introduction

3D point cloud segmentation has wide-ranging applications (Xiao et al., 2024; Ren et al., 2024; Jiang et al., 2024) across various fields. Despite numerous successes in fully supervised learning (Nie et al., 2022; Lai et al., 2022; Zhang et al., 2023b), its effectiveness is constrained by the semantic categories predefined in large-scale, expensive, and fully-annotated datasets (Dai et al., 2017; Armeni et al., 2016). To address this challenge, few-shot 3D point cloud semantic segmentation (**FS-PCS**) has recently attracted increasing attention, enabling models to generalize to unseen/novel categories with just a few annotated samples. Existing FS-PCS methods (Zhao et al., 2021; Xu et al., 2023; Zhu et al., 2023; Mao et al., 2022; Wang et al., 2023; Zhang et al., 2023a; An et al., 2024) typically adhere to the meta-learning framework (Vinyals et al., 2016; Snell et al., 2017; Ren et al., 2018) to transfer knowledge from annotated support point clouds to query point clouds for segmenting novel classes.

However, these methods predominantly focus on unimodal point cloud inputs, overlooking the potential benefits of leveraging multimodal information. Insights from neuroscience (Nanay, 2018; Quiroga et al., 2005; Kuhl & Meltzoff, 1984; Meltzoff & Borton, 1979) suggest that human cognitive learning is inherently multimodal, with different modalities of the same concept exhibiting strong correspondence through the activation of synergistic neurons. Particularly, multimodal signals, such as vision and language, have been shown to play crucial roles, surpassing the performance of only

---

*Corresponding authors: Guolei Sun and Yun Liu

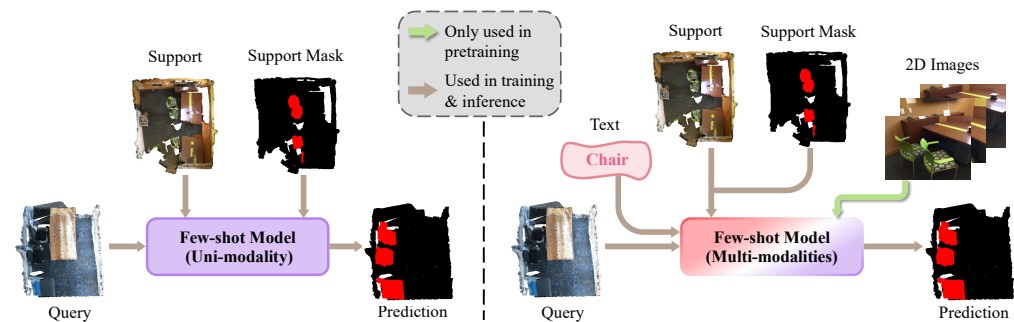

Figure 1: **Comparison between traditional unimodal FS-PCS and our proposed multimodal FS-PCS.** Previous FS-PCS methods only make use of point clouds as unimodal input. In contrast, our proposed model utilizes multimodal information without additional annotation cost to improve FS-PCS by considering the textual modality of class names (*explicit*) and learning simulated features of the 2D modality (*implicit*). During meta-learning and inference, the 2D modality is not needed.

utilizing vision information (Jackendoff, 1987; Smith & Gasser, 2005; Gibson, 1969). In the context of few-shot 3D point cloud semantic segmentation, apart from point cloud modality, additional useful modalities include the corresponding class names and 2D images. Motivated by these important observations, a pertinent question arises: *How can we exploit additional modalities in few-shot 3D point cloud semantic segmentation?*

In this paper, we explore the use of multimodal information in few-shot 3D learning scenarios. Specifically, we propose to incorporate two additional modalities in FS-PCS without additional annotation cost, including the textual modality of category names and the 2D image modality that is usually obtained alongside the capture of 3D point clouds. For the textual modality, it contains condensed semantic information of the object class in the language domain. Since the knowledge of the target class name is typically available during the process of annotating support point clouds, the category name is readily accessible and can be utilized as the textual modality input for free in FS-PCS. For the 2D modality, pairs of 2D images and corresponding 3D point cloud provide dense correspondences between 2D pixels and 3D points, enabling the enhancement of 3D visual features by their 2D counterparts. Notably, we only use the 2D modality during pretraining in an *implicit* manner by utilizing 3D features to simulate 2D features. During meta-learning and inference, no 2D images are needed, ensuring that our model remains independent of images from point clouds. We also demonstrate that training on a dataset without 2D images (*e.g.*, S3DIS (Armeni et al., 2016)) can be achieved by employing the feature extraction module pretrained on other datasets (*e.g.*, ScanNet (Dai et al., 2017)). Thus, the multimodal information used by us is cost-free, as shown in Fig. 1.

Under this cost-free multimodal FS-PCS setup, we introduce a novel model, **M**ulti**M**odal **F**ew-**S**hot **S**egNet (**MM-FSS**), to effectively address FS-PCS by harnessing complementary information from different modalities. MM-FSS processes 3D point cloud inputs by a shared 3D backbone with two heads to extract intermodal and unimodal (point cloud) features, respectively. The intermodal features are firstly pretrained to be aligned with the corresponding 2D visual features extracted from vision-language models (VLMs) such as LSeg (Li et al., 2022) using 2D modality. Then, our model can perform few-shot segmentation using intermodal/unimodal features and text embedding extracted from the VLM's text encoder on textual modality. This design enables the flexible application of our model even if there is no 2D modality available. Specifically, we develop a **M**ultimodal **C**orrelation **F**usion (**MCF**) module to effectively fuse correlations computed from different information sources. The following **M**ultimodal **S**emantic **F**usion (**MSF**) module further improves the fused correlations by utilizing semantic guidance from textual modality, *i.e.*, the target classes, to enhance the point-wise multimodal semantic understanding. Additionally, we propose a simple yet effective **T**est-time **A**daptive **C**ross-modal **C**alibration (**TACC**) technique to mitigate training bias inherent in few-shot models (Cheng et al., 2022). This technique adaptively calibrates predictions during test time by measuring an adaptive indicator for each meta sample to achieve better generalization.

We systematically compare our MM-FSS against existing methods (Zhao et al., 2021; He et al., 2023; Ning et al., 2023; An et al., 2024) on S3DIS (Armeni et al., 2016) and ScanNet (Dai et al., 2017) datasets (§4.2), suggesting the significant superiority of MM-FSS across various settings.

With extensive ablation studies (§4.3), we offer further insights into the efficacy of our designs and showcase the benefits of utilizing free modalities for FS-PCS, shedding light on future research.

Our contributions are three-fold. **(i)** We study the value of multimodal information (textual and 2D modality) in FS-PCS by proposing a novel cost-free multimodal FS-PCS setup. To the best of our knowledge, this is the first work to explore multimodality in this domain. **(ii)** We introduce a novel model, MM-FSS, to effectively exploit information from different modalities, which includes multimodal correlation fusion, multimodal semantic fusion, and test-time adaptive cross-modal calibration modules. **(iii)** Extensive experiments are conducted and validate the value of the proposed setup and the efficacy of the proposed method across different few-shot settings. Our work will inspire future research in this field.

## 2 RELATED WORK

### 2.1 FEW-SHOT 3D POINT CLOUD SEGMENTATION

While many prior works have shown success in fully-supervised 3D point cloud segmentation (Lin et al., 2020; Zhou et al., 2021; Ran et al., 2021; Nie et al., 2022; Lai et al., 2022; Park et al., 2022; Zhang et al., 2023b; Wu et al., 2024; Kolodiazhnyi et al., 2024; Wang et al., 2024a; Han et al., 2024), the high labor cost of annotating point clouds has spurred interest in few-shot methods. The pioneering FS-PCS approach by attMPTI (Zhao et al., 2021) introduced label propagation from support to query points in a transductive manner. Subsequent research has focused on bridging the semantic gap between prototypes and query points (He et al., 2023; Ning et al., 2023; Zhu et al., 2023; Zheng et al., 2024; Xiong et al., 2024; Li et al., 2024) and enhancing representation learning (Mao et al., 2022; Wang et al., 2023; Zhang et al., 2023a; Huang et al., 2024; Zhu et al., 2024; An et al., 2024). For instance, PAP (He et al., 2023) converts support prototypes to better align with query features for alleviating the intra-class variation, QGE (Ning et al., 2023) refines support prototypes in two steps by firstly adapting the support background prototypes and secondly optimizing support prototypes holistically, and 2CBR (Zhu et al., 2023) aligns the support and query distributions by rectifying the bias in support based on co-occurrence features. BFG (Mao et al., 2022) enhances the prototypes with global perception through bidirectional feature globalization. CSSMRA (Wang et al., 2023) uses contrastive self-supervision to overcome pretraining biases. SCAT (Zhang et al., 2023a) leverages multi-scale query and support features for exploring detailed relationships. Seg-NN (Zhu et al., 2024) designs hand-crafted filters for extracting dense features in order to alleviate domain gaps between training and inference. Notably, recent work, COSeg (An et al., 2024), highlights two issues in the previous FS-PCS experimental setting, *i.e.*, foreground leakage and sparse point distribution, and introduces a reasonable setting with a new benchmark to facilitate this field.

### 2.2 MULTIMODAL 3D POINT CLOUD SEGMENTATION

As the unimodal 3D point cloud segmentation models (Liu et al., 2019b;a; Hu et al., 2020; Zhang et al., 2022; Wang et al., 2024b; Zhang et al., 2024; Xu et al., 2024b) show huge progress, increasing attention has been put to explore multimodality for further improvements. The most common modality used to help 3D segmentation is the 2D images. Due to its richer texture and appearance features compared to point clouds, many works have achieved better segmentation performance by learning from the two modalities. The first category, fusion-based methods (Su et al., 2018; Krispel et al., 2020; El Madawi et al., 2019; Meyer et al., 2019; Kundu et al., 2020; Zhuang et al., 2021; Maiti et al., 2023; Liu et al., 2023), fuses the semantic features or predictions from 2D images with the corresponding 3D parts to benefit from both modalities. The second category, distillation-based works (Yan et al., 2022; Tang et al., 2023), applies knowledge distillation (Hinton et al., 2015) to train the student branch on 3D unimodality to learn from the fused multimodal features. Besides the 2D modality, the language modality is also utilized for 3D visual perception, enabling models (Rozenberszki et al., 2022; Jatavallabhula et al., 2023; Peng et al., 2023; Ding et al., 2023; Ha & Song, 2023; Chen et al., 2023; Mei et al., 2024; Xu et al., 2024a) to achieve open-vocabulary 3D segmentation by learning 3D features guided by CLIP (Radford et al., 2021) or other 2D vision-language models (Li et al., 2022; Ghiasi et al., 2022). However, these multimodal methods are designed either for fully supervised or open-vocabulary segmentation. When it comes to FS-PCS, prior methods only make use of unimodal point clouds, potentially due to challenges in integrating additional modalities (further discussion in Appendix C). In contrast, we propose MM-FSS to leverage cost-free multimodal information

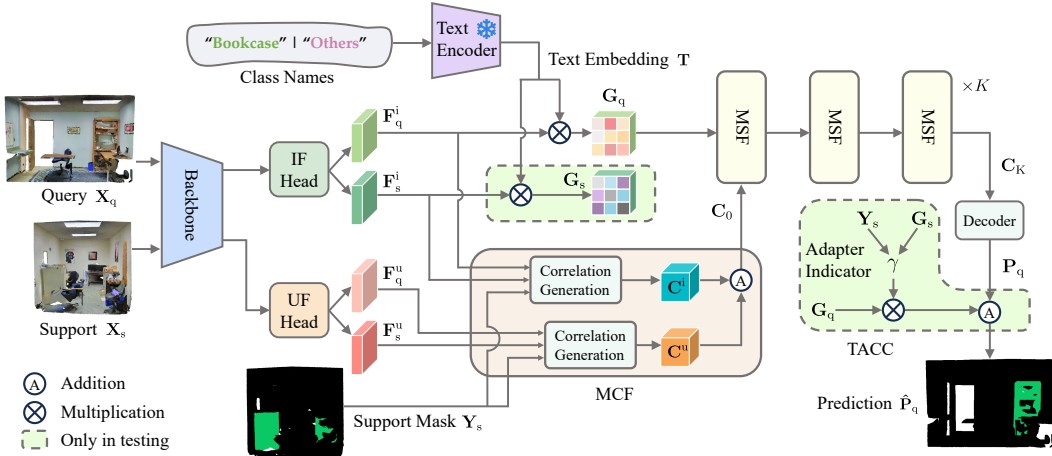

Figure 2: **Overall architecture of the proposed MM-FSS.** Given support and query point clouds, we first generate intermodal features $\mathbf{F}^i_{s/q}$ from the IF head and unimodal features $\mathbf{F}^u_{s/q}$ from the UF head. These features are then forwarded to the MCF module to generate initial multimodal correlations $\mathbf{C}_0$. Moreover, exploiting the alignment between intermodal features $\mathbf{F}^i_q$ and text embeddings $\mathbf{T}$, we use their affinity $\mathbf{G}_q$ as the informative textual semantic guidance to refine the multimodal correlations in the MSF modules. Finally, we propose the TACC, a parameter-free module that adaptively calibrates predictions during test time to effectively mitigate the base bias issue. For clarity, we present the model under the 1-way 1-shot setting.

for improving FS-PCS by fusing textual class names and simulated 2D features. To the best of our knowledge, this is the first study to explore multimodal FS-PCS.

## 3 METHODOLOGY

### 3.1 PROBLEM SETUP

**FS-PCS.** This task can be formulated as the popular episodic paradigm (Vinyals et al., 2016), following prior works (Zhao et al., 2021; An et al., 2024). Each episode corresponds to an $N$-way $K$-shot segmentation task, containing a support set $\mathcal{S} = \left\{ \{\mathbf{X}^{n,k}_s, \mathbf{Y}^{n,k}_s\}^K_{k=1} \right\}^N_{n=1}$ and a query set $\mathcal{Q} = \{\mathbf{X}^n_q, \mathbf{Y}^n_q\}^N_{n=1}$. We use $\mathbf{X}^*_{s/q}$ and $\mathbf{Y}^*_{s/q}$ to denote a point cloud and its corresponding point-level label, respectively. The support set $\mathcal{S}$ includes the samples for $N$ target classes, with each class $n$ described by a $K$-shot group $\{\mathbf{X}^{n,k}_s, \mathbf{Y}^{n,k}_s\}^K_{k=1}$, containing the exclusive labels for that semantic class. The goal of FS-PCS is to segment the query samples $\{\mathbf{X}^n_q\}^N_{n=1}$ into $N$ target classes and 'background' by leveraging the knowledge of the $N$ novel classes from support samples in $\mathcal{S}$.

**Multimodal FS-PCS.** Different from the existing setup, we propose a multimodal FS-PCS setup where two additional modalities exist: the textual modality and the 2D image modality. Formally, for the episode introduced above, we additionally have $N$ class names for $\mathcal{S}$, *e.g.*, '*chair*', '*table*', '*wall*', *etc*. For the 2D image modality, we have 2D RGB images accompanying 3D point clouds during pretraining, but 2D images are not required during meta-learning and inference. In the following discussions, unless stated otherwise, we focus on the 1-way 1-shot setting for clarity. The support and query sets are represented as $\mathcal{S} = \{\mathbf{X}_s, \mathbf{Y}_s\}$ and $\mathcal{Q} = \{\mathbf{X}_q, \mathbf{Y}_q\}$, respectively.

### 3.2 METHOD OVERVIEW

**Our Idea.** Since existing FS-PCS datasets containing three modalities (3D point clouds, class names, and 2D RGB images) are generally on a small scale, it is difficult to directly train models to learn meaningful representations of these modalities. Inspired by the rapid advancements in vision-language models (VLMs), we propose to leverage existing VLMs such as LSeg (Li et al., 2022) and OpenSeg (Ghiasi et al., 2022) to exploit additional modalities for FS-PCS.

Specifically, we adopt the pretrained text encoder of LSeg (Li et al., 2022) to extract text embeddings for class names. These powerful text embeddings provide additional guidance for learning FS-PCS, which is supplementary to the visual guidance extracted from the support set. To utilize the potentially available 2D modality, we propose to use the visual encoder of LSeg to generate 2D visual features, which exhibit excellent generalizability since the LSeg model is pretrained on large-scale 2D datasets. Considering that the 2D modality is not always available for all FS-PCS datasets (Armeni et al., 2016), we employ the extracted 2D features to supervise the learning of 3D point cloud features during pretraining, effectively using 3D features to simulate 2D features. The learned features are referred to as *intermodal features* since they are aware of both 3D and 2D information. This design offers two key advantages: **i)** Our model uses 2D modality in an *implicit* manner and does not require it as input during meta-learning and inference; **ii)** Since the learned intermodal features are aligned with LSeg's 2D visual features, they are therefore aligned with text embeddings. This alignment provides important guidance for subsequent stages, which will be explained in detail later.

**Method Overview.** The overall architecture of the proposed MM-FSS is depicted in Fig. 2. Given support and query point clouds, we first generate two sets of high-level features: intermodal features from the **I**ntermodal **F**eature (**IF**) head and unimodal features (point cloud modality) from the **U**nimodal **F**eature (**UF**) head. Both intermodal and unimodal features are then forwarded to the **M**ultimodal **C**orrelation **F**usion (**MCF**) module to produce multimodal correlations between support and query point clouds. Beyond mining visual connections, we use the LSeg text encoder (Li et al., 2022) to generate text embeddings for class names. We then exploit useful semantic guidance from the textual modality to refine the multimodal correlations in the **M**ultimodal **S**emantic **F**usion (**MSF**) module. During inference, to mitigate training bias (Cheng et al., 2022), we further propose **T**est-time **A**daptive **C**ross-modal **C**alibration (**TACC**) to generate better predictions for novel classes.

Existing FS-PCS approaches (An et al., 2024; Zhao et al., 2021) typically have two training steps: a pretraining step for obtaining an effective feature extractor, and a meta-learning step towards few-shot segmentation tasks. Our method follows this two-step training paradigm. First, we pretrain the backbone and IF head using 3D point clouds and 2D images. Second, we conduct meta-learning to train the model end-to-end while freezing the backbone and IF head. Further training details are provided in Appendix B. In the following, we elaborate on feature extractors (3D backbone, IF and UF heads, text encoder) as well as MCF, MSF, and TACC modules.

### 3.3 FEATURE EXTRACTORS

**Visual Features.** Our method processes point cloud inputs through a joint backbone and two distinct heads of IF and UF, as depicted in Fig. 2. The IF head extracts intermodal features that are aligned with 2D visual features by exploiting the 2D modality, while the UF head focuses solely on 3D point cloud modality. Given the support/query point cloud $\mathbf{X}_{s/q}$, we utilize a shared backbone $\Phi$ to obtain general support features $\mathbf{F}_s = \Phi(\mathbf{X}_s) \in \mathbb{R}^{N_S \times D}$ and query features $\mathbf{F}_q = \Phi(\mathbf{X}_q) \in \mathbb{R}^{N_Q \times D}$, where $D$ is the channel dimension, $N_S$ and $N_Q$ are the respective point counts in $\mathbf{X}_s$ and $\mathbf{X}_q$. Subsequently, these features are processed by the IF head ($\mathcal{H}_{\mathrm{IF}}$) and the UF head ($\mathcal{H}_{\mathrm{UF}}$) to generate intermodal and unimodal features for both support and query inputs, given by:

$$
\begin{aligned}
\mathbf{F}_s^i = \mathcal{H}_{\mathrm{IF}}(\mathbf{F}_s) \in \mathbb{R}^{N_S \times D_t}, \ \mathbf{F}_s^u = \mathcal{H}_{\mathrm{UF}}(\mathbf{F}_s) \in \mathbb{R}^{N_S \times D}, \\
\mathbf{F}_q^i = \mathcal{H}_{\mathrm{IF}}(\mathbf{F}_q) \in \mathbb{R}^{N_Q \times D_t}, \ \mathbf{F}_q^u = \mathcal{H}_{\mathrm{UF}}(\mathbf{F}_q) \in \mathbb{R}^{N_Q \times D}.
\end{aligned}
\tag{1}
$$

$D_t$ denotes the channel dimension of intermodal features, which is aligned with the embedding space of LSeg (Li et al., 2022). The resulting $\mathbf{F}_s^i$ and $\mathbf{F}_s^u$ represent the intermodal and unimodal features for the support point cloud, respectively. $\mathbf{F}_q^i$ and $\mathbf{F}_q^u$ serve the same purpose for the query point cloud.

As mentioned above, the intermodal features, $\mathbf{F}_s^i$ and $\mathbf{F}_q^i$, are specifically trained in the first step to align with 2D visual features from the visual encoder of VLMs (Li et al., 2022; Ghiasi et al., 2022). Following Peng et al. (2023), we employ a cosine similarity loss to minimize the distance between 3D point intermodal features and corresponding 2D pixel features (see Appendix B). Once this step finishes, we fix the backbone and IF head to keep the intermodal features for providing critical guidance for FS-PCS. Then, we start meta-learning end-to-end to fully exploit the intermodal and unimodal features along with text embeddings in conducting FS-PCS.

**Text Embeddings.** We compute embeddings for the 'background' and target classes using the LSeg (Li et al., 2022) text encoder, denoted as $\mathbf{T} = \{\mathbf{t}_0, \cdots, \mathbf{t}_N\} \in \mathbb{R}^{N_C \times D_t}$, where $\mathbf{t}_0$ represents

the text embedding of the 'background' class, and the others represent the text embeddings of the target classes. Here, $N_C = N + 1$ denotes the number of all classes in the $N$-way setting.

## 3.4 CROSS-MODAL INFORMATION FUSION

We have intermodal and unimodal features for support/query point clouds and text embeddings for target classes. Our goal is to predict the segmentation mask of the query point cloud using all available information from different modalities. As in Min et al. (2021), Hong et al. (2022), and An et al. (2024), the core of few-shot segmentation is to build informative correlations between query and support point clouds. To this end, we propose two novel modules for cross-modal knowledge fusion: MCF and MSF. The former integrates intermodal and unimodal features to generate multimodal correlations. The latter exploits the textual semantic guidance to further refine the correlations. The details of these two modules are explained below.

**Multimodal Correlation Fusion.** Contrary to traditional FS-PCS models that rely solely on unimodal inputs (Zhao et al., 2021; He et al., 2023; An et al., 2024), our method calculates multimodal correlations by integrating the two correlations from intermodal and unimodal features. Initially, foreground and background prototypes are generated from the annotated support points for both $\mathbf{F}_s^i$ and $\mathbf{F}_s^u$ using farthest point sampling and points-to-samples clustering, as described in An et al. (2024) and Zhao et al. (2021). This prototype generation, denoted as $\mathcal{F}_{\text{proto}}$, results in:

$$
\begin{aligned}
\mathbf{P}_{\text{fg}}^i, \mathbf{P}_{\text{bg}}^i = \mathcal{F}_{\text{proto}}(\mathbf{F}_s^i, \mathbf{Y}_s, \mathbf{L}_s), \quad & \mathbf{P}_{\text{fg}}^i, \mathbf{P}_{\text{bg}}^i \in \mathbb{R}^{N_P \times D_t}, \\
\mathbf{P}_{\text{fg}}^u, \mathbf{P}_{\text{bg}}^u = \mathcal{F}_{\text{proto}}(\mathbf{F}_s^u, \mathbf{Y}_s, \mathbf{L}_s), \quad & \mathbf{P}_{\text{fg}}^u, \mathbf{P}_{\text{bg}}^u \in \mathbb{R}^{N_P \times D},
\end{aligned}
\tag{2}
$$

where $\mathbf{L}_s$ represents the 3D coordinates of support points, and $N_P$ is the number of prototypes. These prototypes are then concatenated to obtain: $\mathbf{P}_{\text{proto}}^i = \mathbf{P}_{\text{fg}}^i \oplus \mathbf{P}_{\text{bg}}^i \in \mathbb{R}^{(N_C \times N_P) \times D_t}$ and $\mathbf{P}_{\text{proto}}^u = \mathbf{P}_{\text{fg}}^u \oplus \mathbf{P}_{\text{bg}}^u \in \mathbb{R}^{(N_C \times N_P) \times D}$. Subsequently, we calculate the correlations between the query points and these prototypes:

$$
\mathbf{C}^i = \frac{\mathbf{F}_q^i \cdot \mathbf{P}_{\text{proto}}^{i\mathsf{T}}}{\left\|\mathbf{F}_q^i\right\| \left\|\mathbf{P}_{\text{proto}}^{i\mathsf{T}}\right\|}, \quad \mathbf{C}^u = \frac{\mathbf{F}_q^u \cdot \mathbf{P}_{\text{proto}}^{u\mathsf{T}}}{\left\|\mathbf{F}_q^u\right\| \left\|\mathbf{P}_{\text{proto}}^{u\mathsf{T}}\right\|},
\tag{3}
$$

yielding $\mathbf{C}^i \in \mathbb{R}^{N_Q \times (N_C \times N_P)}$ and $\mathbf{C}^u \in \mathbb{R}^{N_Q \times (N_C \times N_P)}$, which represent the point-category relationships between query points and support prototypes within the *intermodal* and *unimodal* *feature spaces*, respectively. This process is termed *correlation generation* in Fig. 2. Next, our MCF module transforms these correlations using two linear layers and then combines them to obtain the aggregated multimodal correlation $\mathbf{C}_0$, as follows:

$$
\mathbf{C}_0 = \mathcal{F}_{\text{lin}}(\mathbf{C}^i) + \mathcal{F}_{\text{lin}}(\mathbf{C}^u), \quad \mathbf{C}_0 \in \mathbb{R}^{N_Q \times N_C \times D},
\tag{4}
$$

where $\mathcal{F}_{\text{lin}}$ represents the linear layer projecting the $N_P$ channels in $\mathbf{C}^{i/u}$ to $D$. The MCF module effectively aggregates point-to-prototype relationships informed by different modalities, enhancing the correlation $\mathbf{C}_0$ with a comprehensive multimodal understanding of the connections between query points and support classes. This enriched understanding facilitates knowledge transfer from support to query point cloud, improving query segmentation.

**Multimodal Semantic Fusion.** While the MCF module effectively merges correlations from different information sources, the semantic information of text embeddings remains untouched, which could provide valuable semantic guidance to improve the correlations. Therefore, we propose the MSF module, as illustrated in Fig. 2. MSF integrates semantic information from text embeddings to refine the correlation output of MCF. Additionally, since the relative importance of visual and textual modalities varies across different points and classes (Yin et al., 2021; Cheng et al., 2021), MSF dynamically assigns different weights to the textual semantic guidance for each query point and target class, accounting for the varying importance between modalities.

Given text embeddings $\mathbf{T}$ and intermodal features $\mathbf{F}_q^i$ of the query point cloud, since the intermodal features $\mathbf{F}_q^i$ are pretrained to simulate the 2D visual features from VLMs (Li et al., 2022), $\mathbf{F}_q^i$ is well-aligned with text embeddings $\mathbf{T}$, and the affinities between them provide informative guidance on how well the query points relate to the target classes. Therefore, we first compute the similarity between the query intermodal features and text embeddings to generate semantic guidance $\mathbf{G}_q \in \mathbb{R}^{N_Q \times N_C}$

for segmenting the target classes, given by:

$$\mathbf{G}_q = \mathbf{F}_q^i \cdot \mathbf{T}^\mathsf{T}. \tag{5}$$

Then, our MSF module consists of $K$ MSF blocks, with the correlation input to the current block denoted as $\mathbf{C}_k$ ($k \in \{0, 1, \cdots, K-1\}$). In each block, point-category weights to consider varying importance between visual and textual modalities are dynamically computed as follows:

$$\mathbf{W}_q = \mathcal{F}_{\mathrm{mlp}}(\mathcal{F}_{\mathrm{expand}}(\mathbf{G}_q) \oplus \mathbf{C}_k), \quad \mathbf{W}_q \in \mathbb{R}^{N_Q \times N_C \times 1}, \tag{6}$$

where $\mathcal{F}_{\mathrm{expand}}$ expands and repeats on the last dimension of $\mathbf{G}_q$, transforming it to $\mathbb{R}^{N_Q \times N_C \times D}$, and $\mathcal{F}_{\mathrm{mlp}}$ represents a multilayer perceptron (MLP). Next, the semantic guidance $\mathbf{G}_q$, weighted by $\mathbf{W}_q$, is aggregated into the correlation input $\mathbf{C}_k$. A linear attention layer (Katharopoulos et al., 2020) and a MLP layer are used to further refine the correlations, given by:

$$\mathbf{C}_k' = \mathbf{G}_q \odot \mathbf{W}_q + \mathbf{C}_k, \tag{7}$$

$$\mathbf{C}_{k+1} = \mathcal{F}_{\mathrm{mlp}}(\mathcal{F}_{\mathrm{attention}}(\mathbf{C}_k')), \tag{8}$$

where $\odot$ denotes the Hadamard product and $\mathcal{F}_{\mathrm{attention}}$ represents the linear attention layer. Note that the residual connections after $\mathcal{F}_{\mathrm{attention}}$ and $\mathcal{F}_{\mathrm{mlp}}$ are omitted here for simplicity.

This MSF module fully leverages useful semantic information from textual modality to enhance the correlations between query and support point cloud, helping to determine the best class for query points. Note that it computes the relative importance between visual and textual modalities for all pairs of points and classes, improving the effective integration of textual modality.

**Loss Function.** After the MSF module with $K$ blocks, the refined correlation $\mathbf{C}_K$ is transformed into the prediction $\mathbf{P}_q \in \mathbb{R}^{N_Q \times N_C}$ by a decoder comprising a KPConv (Thomas et al., 2019) layer and a MLP layer. The whole model is optimized end-to-end by computing cross-entropy loss between the prediction $\mathbf{P}_q$ and the ground-truth label $\mathbf{Y}_q$ for the query point cloud.

### 3.5 Test-time Adaptive Cross-modal Calibration

Few-shot models inevitably introduce a bias towards base classes due to full supervision on these classes during training (Lang et al., 2022; Cheng et al., 2022; Wang et al., 2023; An et al., 2024). When the few-shot model is evaluated on novel classes, this base bias leads to *false activations for base classes existing in test scenes*, impairing generalization.

To mitigate it, we propose a simple yet effective TACC module, exclusively employed during test time. The TACC module exploits the semantic guidance $\mathbf{G}_q$ to calibrate the prediction $\mathbf{P}_q$. Notably, $\mathbf{G}_q$ is derived from the query intermodal features and text embeddings, which are not updated throughout the meta-learning process. Thus, $\mathbf{G}_q$ includes much less bias towards the training categories. Furthermore, $\mathbf{G}_q$ contains rich semantic information for the query point cloud, and $\mathbf{G}_q[i, :]$ represents the probability of assigning $i^{th}$ point to target classes. Building upon this, we propose an adaptive combination of $\mathbf{G}_q$ and $\mathbf{P}_q$ through an adaptive indicator $\gamma$, enabling an appropriate utilization of the semantics in $\mathbf{G}_q$ in the final prediction:

$$\hat{\mathbf{P}}_q = \gamma \mathbf{G}_q + \mathbf{P}_q. \tag{9}$$

Here, $\gamma$ is an adaptive indicator reflecting the quality of semantics contained in $\mathbf{G}_q$. If $\gamma$ is high, the quality of $\mathbf{G}_q$ is good, and more information in $\mathbf{G}_q$ is used. If $\gamma$ is low, the quality of $\mathbf{G}_q$ is unsatisfactory, and less information in $\mathbf{G}_q$ is employed.

**Adaptive Indicator.** The proposed adaptive indicator $\gamma$ is dynamically calculated for each few-shot episode by evaluating $\mathbf{G}_s$ for support samples. Using the support intermodal features $\mathbf{F}_s^i$ and the text embeddings $\mathbf{T}$, we compute $\mathbf{G}_s$, which is then used to generate predicted labels $\mathbf{P}_s$. With the available support labels $\mathbf{Y}_s$ in each episode, the quality of $\mathbf{G}_s$ is quantified by comparing the predicted labels $\mathbf{P}_s$ to $\mathbf{Y}_s$ using the Intersection-over-Union (IoU) score. Since $\mathbf{G}_q$ and $\mathbf{G}_s$ are computed in the same way using intermodal features and text embeddings, this score serves as $\gamma$, indicating the reliability of the semantic guidance in $\mathbf{G}_q$:

$$\gamma = \frac{\sum_i \mathbf{1}_{\{\mathbf{P}_s(i)=1 \wedge \mathbf{Y}_s(i)=1\}}}{\sum_i \mathbf{1}_{\{\mathbf{P}_s(i)=1 \vee \mathbf{Y}_s(i)=1\}}}, \quad \mathbf{P}_s[i] = \arg\max(\mathbf{G}_s[i, :]), \quad \mathbf{G}_s = \mathbf{F}_s^i \cdot \mathbf{T}^\mathsf{T}, \tag{10}$$

| Methods | 1-way 1-shot | | | 1-way 5-shot | | | 2-way 1-shot | | | 2-way 5-shot | | |
|---|---|---|---|---|---|---|---|---|---|---|---|---|
| | $S^0$ | $S^1$ | Mean | $S^0$ | $S^1$ | Mean | $S^0$ | $S^1$ | Mean | $S^0$ | $S^1$ | Mean |
| AttMPTI (Zhao et al., 2021) | 36.32 | 38.36 | 37.34 | 46.71 | 42.70 | 44.71 | 31.09 | 29.62 | 30.36 | 39.53 | 32.62 | 36.08 |
| QGE (Ning et al., 2023) | 41.69 | 39.09 | 40.39 | 50.59 | 46.41 | 48.50 | 33.45 | 30.95 | 32.20 | 40.53 | 36.13 | 38.33 |
| QGPA (He et al., 2023) | 35.50 | 35.83 | 35.67 | 38.07 | 39.70 | 38.89 | 25.52 | 26.26 | 25.89 | 30.22 | 32.41 | 31.32 |
| COSeg (An et al., 2024) | 46.31 | 48.10 | 47.21 | 51.40 | 48.68 | 50.04 | 37.44 | 36.45 | 36.95 | 42.27 | 38.45 | 40.36 |
| COSeg† (An et al., 2024) | 47.17 | 48.37 | 47.77 | 50.93 | 49.88 | 50.41 | 37.15 | 38.99 | 38.07 | 42.73 | 40.25 | 41.49 |
| MM-FSS (ours) | **49.84** | **54.33** | **52.09**(+4.3) | **51.95** | **56.46** | **54.21**(+3.8) | **41.98** | **46.61** | **44.30**(+6.2) | **46.02** | **54.29** | **50.16**(+8.7) |

Table 1: **Quantitative comparison with previous methods in mIoU (%) on the S3DIS dataset.** There are four few-shot settings: 1/2-way 1/5-shot. $S^0/S^1$ refers to using the split 0/1 for evaluation, and 'Mean' represents the average mIoU on both splits. The best results are highlighted in **bold**.

| Methods | 1-way 1-shot | | | 1-way 5-shot | | | 2-way 1-shot | | | 2-way 5-shot | | |
|---|---|---|---|---|---|---|---|---|---|---|---|---|
| | $S^0$ | $S^1$ | mean | $S^0$ | $S^1$ | Mean | $S^0$ | $S^1$ | Mean | $S^0$ | $S^1$ | Mean |
| AttMPTI (Zhao et al., 2021) | 34.03 | 30.97 | 32.50 | 39.09 | 37.15 | 38.12 | 25.99 | 23.88 | 24.94 | 30.41 | 27.35 | 28.88 |
| QGE (Ning et al., 2023) | 37.38 | 33.02 | 35.20 | 45.08 | 41.89 | 43.49 | 26.85 | 25.17 | 26.01 | 28.35 | 31.49 | 29.92 |
| QGPA (He et al., 2023) | 34.57 | 33.37 | 33.97 | 41.22 | 38.65 | 39.94 | 21.86 | 21.47 | 21.67 | 30.67 | 27.69 | 29.18 |
| COSeg (An et al., 2024) | 41.73 | 41.82 | 41.78 | 48.31 | 44.11 | 46.21 | 28.72 | 28.83 | 28.78 | 35.97 | 33.39 | 34.68 |
| COSeg† (An et al., 2024) | 41.95 | 42.07 | 42.01 | 48.54 | 44.68 | 46.61 | 29.54 | 28.51 | 29.03 | 36.87 | 34.15 | 35.51 |
| MM-FSS (ours) | **46.08** | **43.37** | **44.73**(+2.7) | **54.66** | **45.48** | **50.07**(+3.5) | **43.99** | **34.43** | **39.21**(+10.2) | **48.86** | **39.32** | **44.09**(+8.6) |

Table 2: **Quantitative comparison with previous methods in mIoU (%) on the ScanNet dataset.**

where $\mathbf{1}_{\{x\}}$ is the indicator function that equals one if x is true and zero otherwise, and $\mathbf{P}_s[i]$ denotes the predicted class index for the $i^{th}$ support point. This adaptive indicator ensures that the TACC module effectively mitigates training bias by dynamically calibrating predictions during test time, leading to improved few-shot generalization.

# 4 EXPERIMENTS

## 4.1 EXPERIMENTAL SETUP

**Datasets.** We evaluate our method on two popular FS-PCS datasets: S3DIS (Armeni et al., 2016) and ScanNet (Dai et al., 2017). ScanNet provides 2D RGB images of 3D scenes while S3DIS lacks. The two datasets allow us to demonstrate our model's effectiveness in exploiting multimodal data and its capability to excel in FS-PCS even without 2D images on a given dataset. Our model leverages the 2D modality implicitly, enabling the flexible use of pretrained weights from ScanNet to initiate meta-learning on S3DIS. Following Zhao et al. (2021), we divide the large-scale scenes into 1m × 1m blocks. We adhere to the standard data processing protocol from An et al. (2024), voxelizing raw input points within each block using a 0.02m grid size and uniformly sampling to maintain a maximum of 20,480 points per block.

**Implementation Details.** For our architecture, we employ the first two blocks of Stratified Transformer (Lai et al., 2022) as our backbone, with the IF and UF heads following the design of its third stage. By default, we utilize 2 MSF blocks for S3DIS and 4 MSF blocks for ScanNet. The initial pretraining phase spans 100 epochs, while the subsequent meta-learning phase includes 40,000 episodes, following An et al. (2024). For optimization, we use the AdamW optimizer, setting a weight decay of 0.01 and a learning rate of 0.006 during pretraining. The learning rate is reduced to 0.0001 during the meta-learning phase. As in An et al. (2024), the evaluation sets consist of 1,000 episodes per class in the 1-way setting and 100 episodes per class combination in the 2-way setting.

## 4.2 COMPARISON WITH STATE-OF-THE-ART METHODS

We compare MM-FSS with previous models on the S3DIS (Armeni et al., 2016) and ScanNet (Dai et al., 2017) datasets, detailed in Tab. 1 and Tab. 2, respectively. We also evaluate a variant of the previously leading method COSeg (An et al., 2024), denoted as COSeg†, retrained using the same 2D-aligned pretrained backbone weights as our model. Despite leveraging the 2D-aligned backbone weights, COSeg† does not significantly improve over COSeg, highlighting the critical role of well-designed fusion modules in achieving significant advancements.

| MCF | MSF | 1-shot | 5-shot |
|-----|-----|--------|--------|
|     |     | 40.69  | 45.51  |
| ✓   |     | 41.45  | 46.38  |
|     | ✓   | 42.21  | 46.46  |
| ✓   | ✓   | **42.83** | **48.04** |

(a)

| IF head | UF head | TACC | 1-shot | 5-shot |
|---------|---------|------|--------|--------|
| ✓       |         |      | 35.10  | 37.32  |
|         | ✓       |      | 40.69  | 45.51  |
| ✓       | ✓       |      | 42.83  | 48.04  |
| ✓       | ✓       | ✓    | **44.73** | **50.07** |

(b)

| $K$ | 1-shot | 5-shot |
|-----|--------|--------|
| 3   | 43.33  | 45.97  |
| 4   | 42.83  | 48.04  |
| 5   | **44.69** | **48.36** |

(c)

| 3D | Image | Text | 1-shot | 5-shot |
|----|-------|------|--------|--------|
| ✓  |       |      | 40.69  | 45.51  |
| ✓  | ✓     |      | 41.45  | 46.38  |
| ✓  | ✓     | ✓    | **44.73** | **50.07** |

(d)

|         | $1:0$ | $1:0.5$ | $1:1$ | $0:1$ | $\gamma:1$ |
|---------|-------|---------|-------|-------|------------|
| 1-shot  | 35.10 | 43.24   | 43.76 | 42.83 | **44.73**  |
| 5-shot  | 37.32 | 48.12   | 48.85 | 48.04 | **50.07**  |

(e)

| Methods    | 1-shot | 5-shot |
|------------|--------|--------|
| MSF-linear | 40.80  | 45.79  |
| Default    | **42.83** | **48.04** |

(f)

| Methods | FLOPs | Params |
|---------|-------|--------|
| COSeg (An et al., 2024) | **27.76G** | **7.75M** |
| MM-FSS (ours)           | 29.21G | 10.25M |

(g)

Table 3: **Ablation study.** (a) Effect of fusion modules. (b) Effect of interactions between two feature heads. (c) Impact of the number of MSF layers. (d) Performance gains from each modality. (e) Impact of different coefficients in TACC. (f) Weighting methods in MSF. (g) Complexity analysis.

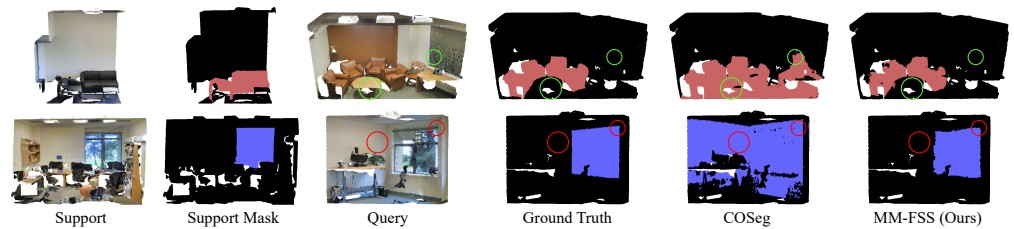

Figure 3: **Qualitative comparison between COSeg and our proposed MM-FSS in the 1-way 1-shot setting on the S3DIS dataset.** The target classes in the first and second rows are sofa and window, respectively. Colored circles highlight regions where predictions from COSeg and MM-FSS differ significantly to facilitate visual comparison.

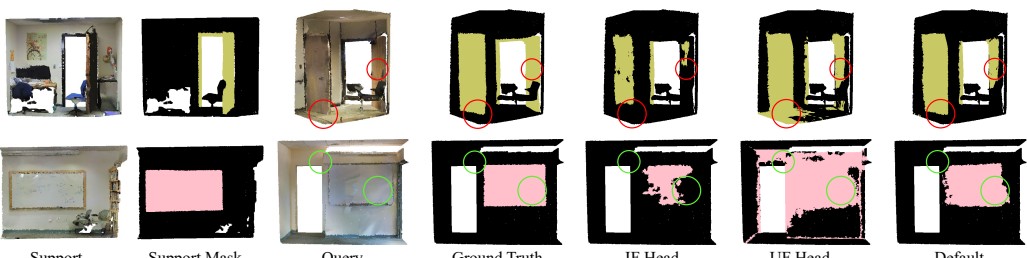

Figure 4: **Qualitative comparison of predictions from each head and our final prediction using TACC (Default) in the 1-way 1-shot setting on the S3DIS dataset.** The target classes in the first and second rows are door and board, respectively.

In contrast, MM-FSS consistently outperforms the former state-of-the-art across all settings, demonstrating superior cross-modal knowledge integration to enhance novel class segmentation. Specifically, on the ScanNet dataset, MM-FSS records average mIoU increases of **+3.41%** in the 1-way and **+9.92%** in the 2-way settings over COSeg. Similarly, it achieves **+4.53%** and **+8.58%** improvements on the S3DIS dataset in the 1/2-way settings, respectively. Visual comparisons in Fig. 3 further illustrate MM-FSS's advanced few-shot segmentation capabilities.

Overall, our model secures average mIoU improvements of **+3.97%** and **+9.25%** across the 1/2-way settings on both datasets. The greater gains in 2-way settings can be attributed to the higher demands on a model's ability to learn novel knowledge under these 2-way conditions. With limited input from support point clouds, models typically struggle to fully learn novel classes for accurate segmentation. However, MM-FSS excels in integrating knowledge from multiple modalities, fostering a deeper comprehension of novel classes. This performance gap underscores our model's superior ability to utilize multimodal knowledge for FS-PCS and the importance of considering commonly-ignored multimodal information to enhance few-shot generalization for future research.

## 4.3 ABLATION STUDY

In this section, unless stated otherwise, we report the mIoU results for both 1-way 1/5-shot settings on ScanNet as the *mean of all splits*.

**Impact of the Fusion Modules.**    Tab. 3a evaluates the effectiveness of our fusion modules. Results show that employing either MCF or MSF individually enhances mIoU, demonstrating their ability to effectively utilize multimodal knowledge. Moreover, combining both MCF and MSF together further improves performance, confirming that their fusion strategies are both essential and complementary for enhancing few-shot learning.

**Ablation on the Feature Heads.**    Tab. 3b examines the interaction effects between the IF and UF heads. Results in the third row indicate that our cross-modal fusion modules effectively combine the capabilities of both heads to learn enhanced multimodal knowledge. Additionally, the TACC module leverages the IF head's semantic guidance to mitigate the UF head's training bias, leading to further mIoU gains, as shown in Fig. 4.

**Impact of the Number of MSF Blocks.**    Tab. 3c showcases the performance of different numbers of MSF blocks, evaluated in the absence of the TACC module. The results demonstrate that increasing the number of MSF blocks enhances few-shot performance. By default, We use 4 MSF blocks for the ScanNet dataset.

**Performance Gains from Each Modality.**    In Tab. 3d, we provide results for different modality combinations to evaluate their respective contributions. Tab. 3d shows that adding the image modality improves the 3D-only baseline, and further incorporating the textual modality leads to better results. This demonstrates our model's effectiveness in fully leveraging the complementary strengths of different modalities for a comprehensive understanding of novel classes.

**Influence of the Coefficients in TACC.**    Tab. 3e assesses how varying coefficients affect prediction calibration in TACC, denoting the coefficients for $\mathbf{G}_q$ and $\mathbf{P}_q$ in Eq. (9) as $a$:$b$. Using only $\mathbf{G}_q$ (1:0) yields the lowest performance due to the IF head's limitations in utilizing support samples for learning novel classes. Fixed coefficients (1:1 and 1:0.5) are unable to dynamically adjust calibration and only slightly improve over the baseline (0:1). Conversely, the adaptive indicator $\gamma$ notably enhances mIoU, proving its superiority in dynamically calibrating predictions for each meta sample.

**Weighting Methods in the MSF Module.**    In MSF, considering the varying relative importance between textual and visual modalities, we dynamically assign weights $\mathbf{W}_q$ in Eq. (7) to the textual semantic guidance across points and classes. Tab. 3f compares the results of using a simple linear combination (denoted as MSF-linear) in Eq. (7) against our detailed weighting method (denoted as Default) within the MSF layers, demonstrating the effectiveness of our proposed weighting approach in appropriately integrating the textual semantic guidance.

**Complexity Analysis.**    Tab. 3g presents a comparison of the FLOPs and parameter count between our model and the previous state-of-the-art method, COSeg (An et al., 2024). The results show that, when achieving significant performance gains, our model incurs only a small increase in computational cost and parameters, demonstrating a superior balance between efficiency and performance.

## 5 CONCLUSION

In this paper, we explore the possibility of exploiting additional modalities for improving FS-PCS. We first propose a novel cost-free multimodal FS-PCS setup by integrating the textual modality of category names and the 2D image modality. Under our cost-free setup, we present MM-FSS, the first multimodal FS-PCS model designed to utilize the textual modality explicitly and 2D modality implicitly to maximize its adaptability across datasets. MM-FSS combines MCF and MSF to effectively aggregate multimodal knowledge, enriching the comprehension of novel concepts from both correlation and semantic perspectives, which are mutually important and complementary. Furthermore, to mitigate the inherent training bias issue in FS-PCS, we introduce the TACC technique, which dynamically calibrates predictions during inference by leveraging semantic guidance from textual modality for each meta sample. MM-FSS achieves significant improvements over existing methods across all settings. Overall, our research provides valuable insights into the importance of commonly-ignored free modalities in FS-PCS and suggests promising directions for future studies.

ACKNOWLEDGMENTS

This research is supported in part by the Agency for Science, Technology, and Research (A*STAR) under its MTC Programmatic Fund (No. M23L7b0021), and the Pioneer Centre for AI, DNRF grant number P1.

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

## A ADDITIONAL EXPERIMENTS

| Methods | 1-way 1-shot | | | 1-way 5-shot | | | 2-way 1-shot | | | 2-way 5-shot | | |
|---|---|---|---|---|---|---|---|---|---|---|---|---|
| | $S^0$ | $S^1$ | mean | $S^0$ | $S^1$ | Mean | $S^0$ | $S^1$ | Mean | $S^0$ | $S^1$ | Mean |
| AttMPTI (Zhao et al., 2021) | 34.03 | 30.97 | 32.50 | 39.09 | 37.15 | 38.12 | 25.99 | 23.88 | 24.94 | 30.41 | 27.35 | 28.88 |
| QGE (Ning et al., 2023) | 37.38 | 33.02 | 35.20 | 45.08 | 41.89 | 43.49 | 26.85 | 25.17 | 26.01 | 28.35 | 31.49 | 29.92 |
| QGPA (He et al., 2023) | 34.57 | 33.37 | 33.97 | 41.22 | 38.65 | 39.94 | 21.86 | 21.47 | 21.67 | 30.67 | 27.69 | 29.18 |
| COSeg (An et al., 2024) | 41.73 | 41.82 | 41.78 | 48.31 | 44.11 | 46.21 | 28.72 | 28.83 | 28.78 | 35.97 | 33.39 | 34.68 |
| COSeg$^\dagger$ (An et al., 2024) | 41.95 | 42.07 | 42.01 | 48.54 | 44.68 | 46.61 | 29.54 | 28.51 | 29.03 | 36.87 | 34.15 | 35.51 |
| MM-FSS (LSeg) | **46.08** | **43.37** | **44.73** | **54.66** | **45.48** | **50.07** | **43.99** | **34.43** | **39.21** | **48.86** | **39.32** | **44.09** |
| MM-FSS (OpenSeg) | **43.74** | **43.19** | **43.47** | **50.60** | **47.09** | **48.85** | **37.80** | **35.40** | **36.60** | **44.31** | **38.48** | **41.40** |

Table 4: **Quantitative comparison with previous methods in terms of mIoU (%) on the ScanNet dataset.** The last two rows represent the FS-PCS performance of our model using different 2D VLMs (LSeg (Li et al., 2022) and OpenSeg (Ghiasi et al., 2022)) in pretraining.

| Aggregation | $S^0$ | $S^1$ | mean |
|---|---|---|---|
| mean | 55.46 | 44.55 | 50.01 |
| max | 54.66 | 45.48 | 50.07 |
| min | 54.00 | 43.48 | 48.74 |

Table 5: Different choices to aggregate adaptive indicator values in the 5-shot setting.

| Weights | 1-shot | 5-shot |
|---|---|---|
| 0.5 : 1 | 41.00 | 45.98 |
| 1 : 1 | 41.45 | 46.38 |

Table 6: Different weights for the MCF module.

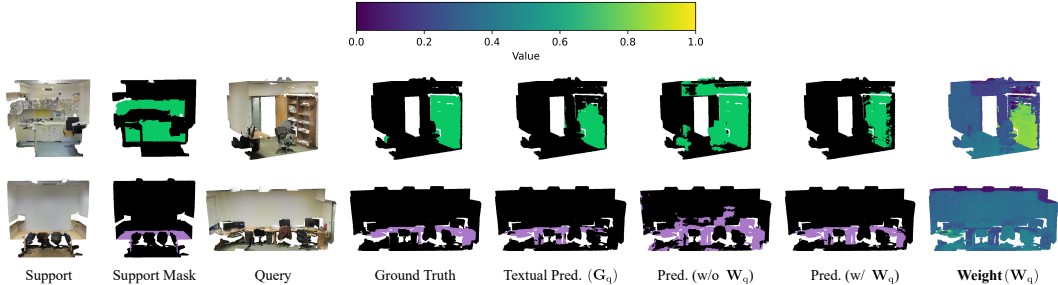

Figure 5: Visualization on the effects of weight $\mathbf{W}_q$ between textual and visual modalities in Eq. (7). The last column displays the heatmap of $\mathbf{W}_q$ with the color bar referenced at the top. Higher values indicate larger weights assigned to textual guidance $\mathbf{G}_q$. Each row represents the 1-way 1-shot setting on the S3DIS dataset targeting bookcase and table, respectively, arranged from top to bottom.

**Ablation study on using different vision-language models (VLMs) in pretraining.** In the initial training phase of our model, we pretrain the backbone and IF head using both 3D point clouds and 2D images. The 2D modality is implicitly incorporated by learning from 2D features extracted by existing VLMs. In §4, we demonstrate the superior performance of our model pretrained with the well-known VLM LSeg (Li et al., 2022). Here, we further investigate the effects of using another VLM, OpenSeg (Ghiasi et al., 2022), for pretraining. The results in the last row of Tab. 4 show that our model pretrained with OpenSeg still outperforms prior methods by a significant margin across all few-shot settings. Additionally, the overall performance of MM-FSS (OpenSeg) is comparable to that of MM-FSS (LSeg), and in some few-shot settings, such as the 1-way 5-shot setting of split 1, it can perform better. These results underscore the superior robustness and generalizability of our method in learning and harnessing the 2D modality across diverse pretraining sources to effectively address FS-PCS.

**Ablation study on the computation of adaptive indicator in the 5-shot setting.** In this ablation study, we compare different aggregation methods of computing the adaptive indicator $\gamma$ in the 5-shot context. For each support point cloud sample, we can compute one value of $\gamma$ following Eq. (10). Thus, in the 5-shot setting, we obtain 5 values of $\gamma$ from each shot, which can be aggregated into a single value using mean, max, or min operations. We evaluate the performance effects of using

these aggregation operations under the 1-way 5-shot settings in Tab. 5, including mIoU results on splits 0 and 1 of ScanNet. Both mean and max aggregation yield comparable performance, while the min aggregation results in approximately a 1.3% decrease in performance compared to the mean and max operations. This indicates that the min operation is less effective in reliably capturing the overall semantic quality of the cross-modality guidance $\mathbf{G}_q$. By default, we employ max aggregation for the 5-shot setting.

**Ablation study on employing different weights for the MCF module.** The MCF module is designed to generate multimodal correlations by leveraging cross-modal knowledge. It fuses two types of correlations derived from intermodal and unimodal features for new multimodal correlations. In Tab. 6, we present the performance in the 1-way 1/5-shot settings of using different weights for fusing these correlations as per Eq. (4). The notation $a : b$ in Tab. 6 indicates the respective weights applied to $\mathcal{F}_{\mathrm{lin}}(\mathbf{C}^{\mathrm{i}})$ and $\mathcal{F}_{\mathrm{lin}}(\mathbf{C}^{\mathrm{u}})$ before their summation. The results show minimal performance variance between the ratios $1 : 1$ and $0.5 : 1$, demonstrating the robustness of this module in learning to capture useful cross-modal information for enhanced multimodal correlations.

**Visualization on the Effects of Weight $\mathbf{W}_q$.** We analyze the effects of the weight $\mathbf{W}_q$ between textual and visual modalities as specified in Eq. (7). When target classes *differ substantially in shape or appearance* between support and query samples, textual guidance becomes crucial. In such cases, visual correlations alone struggle to establish meaningful connections, and the model relies more on textual guidance for better segmentation. In Fig. 5, the first row shows a *bookcase* target class with notable visual differences between support and query. Here, visual correlations alone ($6^{th}$ column) are insufficient, and $\mathbf{W}_q$ assigns higher weights ($8^{th}$ column) to regions highlighted by textual guidance ($5^{th}$ column), leading to improved predictions ($7^{th}$ column). In contrast, the second row shows a *table* target class visually similar in both support and query. Here, $\mathbf{W}_q$ is *more evenly distributed* across the query points, balancing contributions from both visual and textual sources.

# B    ADDITIONAL IMPLEMENTATION DETAILS

**Training Strategy.** We provide more details on our training strategy. Our proposed model is designed as a unified architecture with two heads sharing the same backbone network. The Intermodal Feature (IF) head generates intermodal features, while the Unimodal Feature (UF) head focuses solely on features from the point cloud modality. Effective training for extracting informative intermodal and unimodal features is crucial for achieving optimal performance. Simultaneously training both heads might complicate and destabilize the optimization process due to significant heterogeneity across different modalities (Morency & Baltrušaitis, 2017; Lu et al., 2023) and distinct supervision objectives. Furthermore, existing cross-modal models (Peng et al., 2023) are typically trained under standard paradigms, and transferring such cross-modality alignment learning for the IF head into the episodic training paradigm can impact performance. Hence, we adopt a two-step training strategy to mitigate potential performance issues.

**Pretraining Details.** In the first step, we concentrate on training the IF head to learn robust 3D features aligned with 2D modality, providing a solid foundation for subsequent episodic training. Specifically, given the 3D coordinates of a point $\mathbf{p} \in \mathbb{R}^3$ in a point cloud and an RGB image $\mathbf{I}$ with resolution of $H \times W$ for the scene, we align the 3D point $\mathbf{p}$ to its corresponding 2D pixel $\mathbf{u} = (u, v)$ on the image plane through the projection $\tilde{\mathbf{u}} = M_{int} \cdot M_{ext} \cdot \tilde{\mathbf{p}}$, where $M_{int}$ is the camera-to-pixel intrinsic matrix, $M_{ext}$ is the world-to-camera extrinsic matrix, and $\tilde{\mathbf{u}}$ and $\tilde{\mathbf{p}}$ are the homogeneous coordinates of $\mathbf{u}$ and $\mathbf{p}$, respectively.

The 2D features $\mathbf{F}_{2\mathrm{D}} \in \mathbb{R}^{H \times W \times D_t}$ aligned with text modality can be extracted using the pretrained image encoder in LSeg (Li et al., 2022) or other VLMs (Ghiasi et al., 2022), and the 3D features $\mathbf{F}_{3\mathrm{D}} \in \mathbb{R}^{M \times D_t}$ of the point cloud with $M$ points are derived from the IF head. Then, for matched 3D points and 2D pixels from the 2D-3D correspondences, we optimize the backbone and IF head using a cosine similarity loss to ensure close alignment between the 3D point features from $\mathbf{F}_{3\mathrm{D}}$ and their paired 2D pixel features in $\mathbf{F}_{2\mathrm{D}}$, following Peng et al. (2023).

Once the IF head and backbone are trained, they are frozen during the subsequent episodic training phase to maintain the integrity of the learned intermodal features. Therefore, we ensure that the expressive intermodal features from IF head are preserved and ready for cross-modality integration within our proposed fusion modules during episodic training.

For datasets like ScanNet (Dai et al., 2017), which provide 2D images and camera matrices, direct feature alignment is feasible. For datasets without 2D images, such as S3DIS (Armeni et al., 2016), we can directly use the pretrained IF head and backbone from ScanNet. The pretraining step is to align with the VLMs embedding space *without using any semantic labels*, making the pretrained weights *class-agnostic, generic, and transferable*. This allows us to directly employ pretrained weights from 2D-3D datasets for starting meta-learning on 3D-only datasets.

**Model Details.** Following An et al. (2024), the Stratified Transformer (Lai et al., 2022) serves as our backbone on both S3DIS and ScanNet datasets, using the first two blocks from the Stratified Transformer architecture designed for S3DIS. The IF and UF heads are the same as the third block of the same architecture. Features from the backbone and the two heads are at 1/4 and 1/16 of the original point cloud resolution, respectively. For extracting intermodal or unimodal features, we perform interpolation (Qi et al., 2017) to upsample the 1/16 features from the IF or UF head 4× and concatenate them to the 1/4 backbone features. Then, a MLP is applied to the concatenated features to obtain the final intermodal or unimodal features. The channel dimension of unimodal features is 192, and the dimension of intermodal features is aligned with the pretrained VLMs used in the first pretraining step. For LSeg (Li et al., 2022), the dimension is 512, while for OpenSeg (Ghiasi et al., 2022), it is 768. Following An et al. (2024), input features from both datasets include XYZ coordinates and RGB colors. We extract 100 prototypes ($N_P = 100$) per class; for $k$-shot settings with $k > 1$, we sample $N_P/k$ prototypes from each shot and concatenate them to obtain $N_P$ prototypes. Training and inference are conducted on four RTX 3090 GPUs. Our meta-learning and inference adopt the episodic paradigm (Vinyals et al., 2016). The episode construction follows prior works (Zhao et al., 2021; An et al., 2024) where support sets are selected by randomly sampling a target class and choosing point clouds containing that class.

## C  ADDITIONAL DISCUSSIONS

**Challenges of Utilizing Multimodality in Few-shot Segmentation.** In fully supervised or open-vocabulary segmentation (Liu et al., 2023; Peng et al., 2023), the input is *a single point cloud* and the output is the *predicted semantic labels for that input*. The leverage of multimodality in these tasks focuses on enhancing feature representations of the individual input point cloud, which is relatively straightforward to design and implement. In contrast, few-shot segmentation involves *multiple point cloud inputs* (support and query), where the goal is to segment novel classes in the query based on knowledge derived from the support set. The essence of few-shot segmentation lies in mining *meaningful connections between query and support point clouds* (An et al., 2024). Therefore, when incorporating multimodality in few-shot cases, how to effectively leverage multimodal features to *establish informative correlations* and *facilitate the knowledge transfer* from support to query poses unique challenges. To this end, our proposed MCF and MSF modules effectively exploit visual and textual modalities to construct comprehensive multimodal connections, and TACC uses cross-modal information to calibrate predictions, achieving more robust support-to-query knowledge transfer.

**Insights of the MSF module for enhanced correlations.** The MSF module enhances cross-modal visual correlations by incorporating textual semantic guidance, that is *supplementary to the visual guidance* derived from the support set. The detailed process is as follows:

1. MSF computes the affinities between intermodal query features and text embeddings, *producing the textual semantic guidance* $\mathbf{G}_q$ (Eq. (5)). It quantifies how well each query point relates to the target classes *in the text embedding space*, adding semantic context beyond visual correlations.

2. Recognizing that the relative importance of visual and textual modalities varies across points and classes, MSF *introduces point-category weights* $\mathbf{W}_q$. These weights are determined based on information from both visual and textual modalities as in Eq. (6), dynamically weighting their contributions.

3. The weighted textual semantic guidance, $\mathbf{G}_q \odot \mathbf{W}_q$, is then aggregated with the visual correlations (Eq. (7)).

By leveraging supplementary textual guidance and dynamically weighting its contributions for each point and class, the MSF module results in enhanced multimodal correlations, effectively combining

the complementary strengths of textual and visual information to better determine the best class for query points. The visualizations in Fig. 5 further present the detailed effects of MSF.

**Insights on the integration of 2D modality.** The 3D point cloud and 2D image modalities exhibit *distinct yet complementary* characteristics. 3D point clouds provide rich spatial geometry. However, they are inherently unstructured, lacking natural topology, and their sparsity—points distributed irregularly and concentrated on surfaces—limits the representation of fine-grained details (Lai et al., 2022). 2D images, in contrast, offer dense, structured representations encoding texture, color, and details on a pixel grid. However, they lack direct geometric cues about depth or the spatial structure of the scene (Wu et al., 2023). Despite their differences, the two modalities *share a natural correspondence*: a point in the 3D point cloud typically aligns with a pixel in a 2D image captured from the same perspective. This alignment allows combining the two modalities, laying the foundation for leveraging the strengths of the 2D modality to enhance 3D few-shot segmentation.

Our approach leverages these insights to incorporate the 2D modality in an *implicit* manner during pretraining, *requiring no additional semantic labels*. Specifically, we use the visual encoder of VLMs to generate 2D visual features, which supervise 3D features from the IF head to simulate 2D features. Then, the learned 3D features serve as *a source of 2D information*, exploited to build multimodal understanding of novel classes during subsequent meta-learning and inference stages. Further pretraining details can be found in Appendix B.

**Limitations and Broader Impacts.** Though our method significantly outperforms existing methods by exploiting free modalities, it might learn inductive bias towards the studied datasets and its efficacy on other scenarios needs to be studied before deployed in a practical perception system. Since our method needs to be trained on GPUs, the development and potential deployment lead to carbon emissions and have a negative impact on the environment.

## D ADDITIONAL VISUAL RESULTS

In this section, we provide more extensive visual comparisons to underscore the efficacy of our approach in handling few-shot segmentation tasks. In Fig. 6 and Fig. 7, we provide the additional segmentation results that compare MM-FSS with the previously established state-of-the-art model, COSeg (An et al., 2024). These comparisons clearly demonstrate the enhanced few-shot segmentation capabilities of MM-FSS, illustrating its effective integration of different modalities for capturing a more comprehensive understanding of novel concepts.

Moreover, Fig. 8 includes further visual comparisons across the two feature heads in our model—the Intermodal Feature (IF) head and the Unimodal Feature (UF) head—and the final predictions obtained by fusing outputs from both heads through our TACC module. These results illustrate that the IF head is less prone to training bias, in contrast to the UF head which exhibits greater bias due to its optimization during the meta-learning step. Our proposed TACC effectively leverages the bias-resistant properties of intermodal features to calibrate final predictions during test time by dynamically controlling the calibration for each meta sample, greatly improving the few-shot generalization ability.

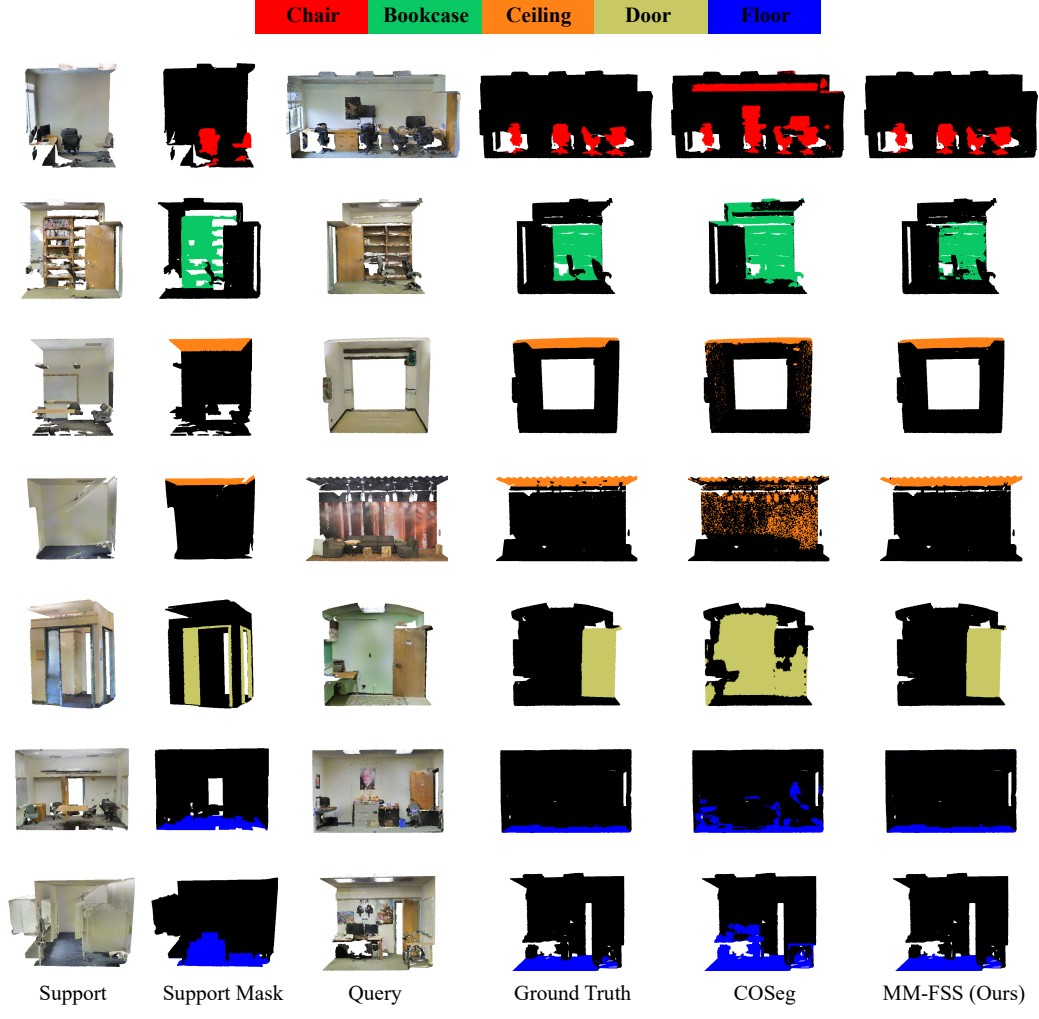

Figure 6: Visual comparison between COSeg (An et al., 2024) and our proposed MM-FSS on the S3DIS dataset. Each row represents one 1-way 1-shot segmentation task with the target class of the given color. MM-FSS predicts masks of higher quality and fewer artifacts compared to COSeg.

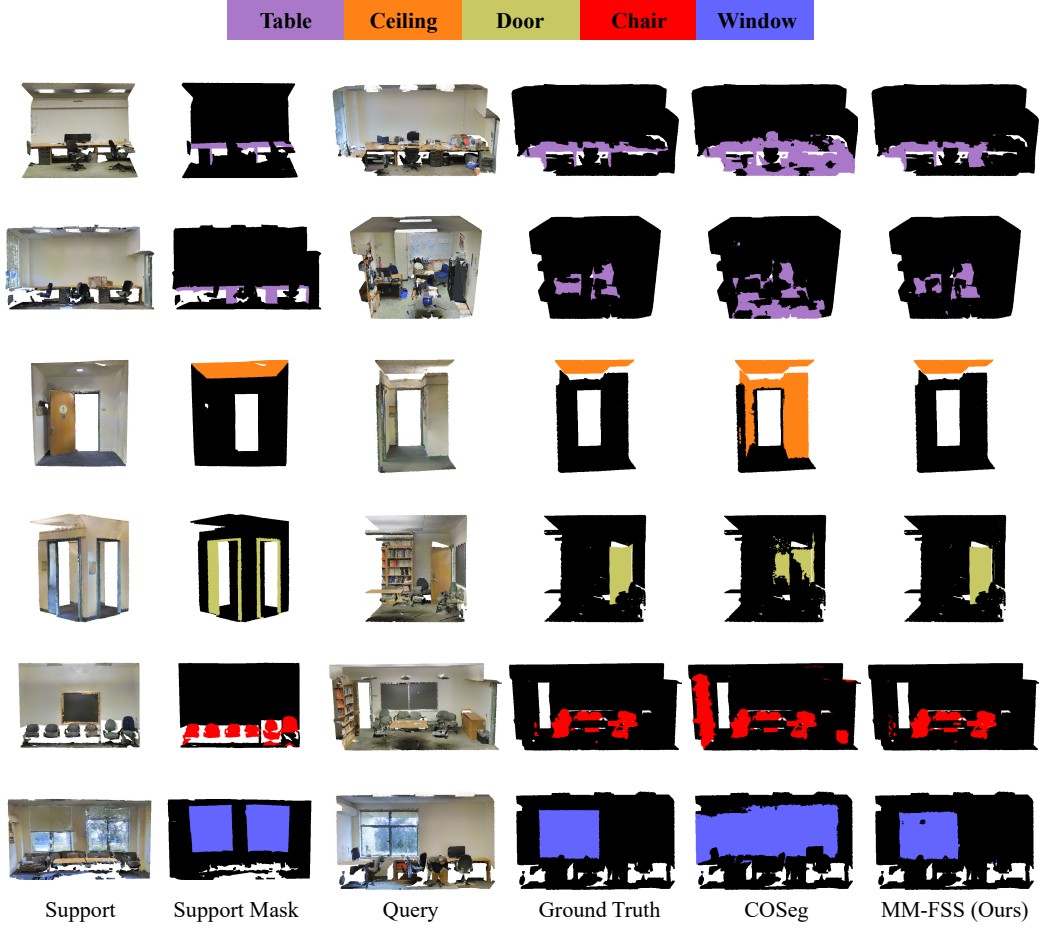

Figure 7: Visual comparison between COSeg (An et al., 2024) and our proposed MM-FSS on the S3DIS dataset. Each row represents one 1-way 1-shot segmentation task with the target class of the given color. MM-FSS predicts masks of higher quality and fewer artifacts compared to COSeg.

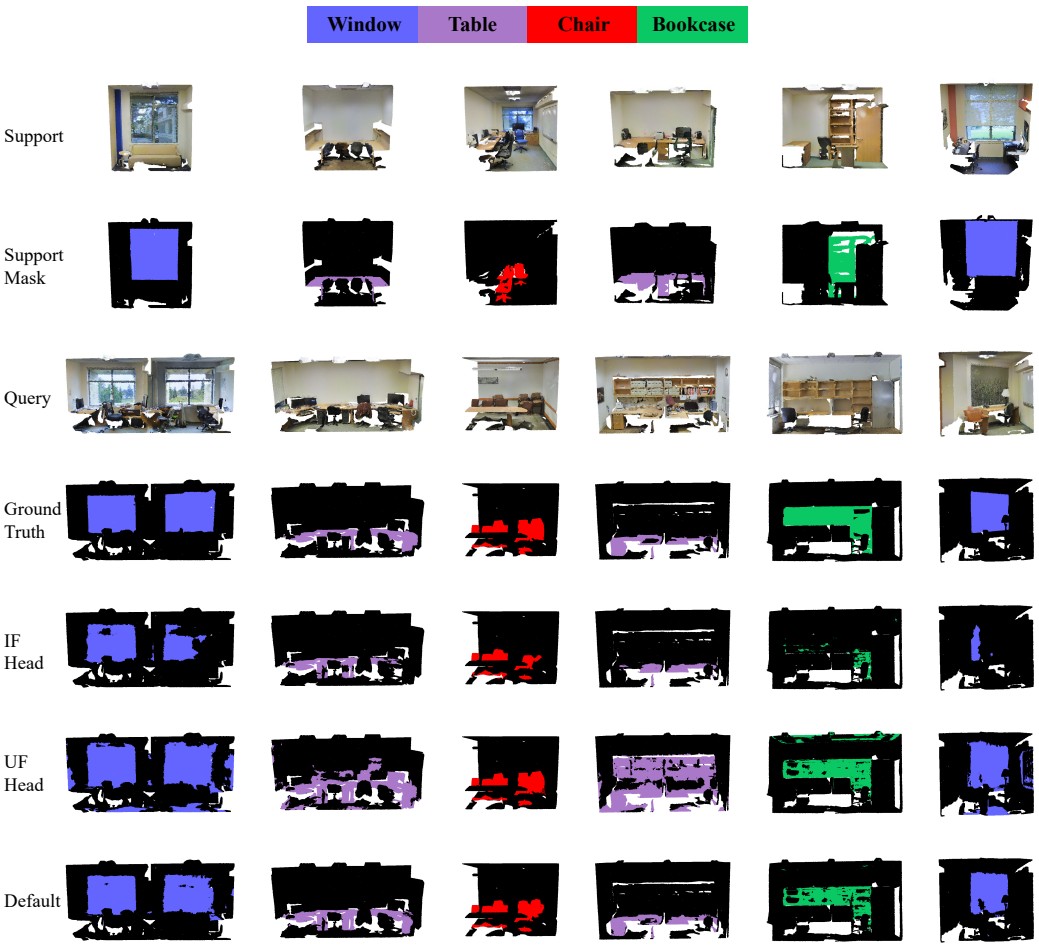

Figure 8: Visual comparison of predictions from each head and our final prediction using TACC (Default) on the S3DIS dataset. Each column represents one 1-way 1-shot segmentation task with the target class of the given color.

