# OpenReview forum: "Multimodality Helps Few-shot 3D Point Cloud Semantic Segmentation"
_ICLR.cc/2025/Conference — ICLR 2025 Spotlight_

### Official Review · Reviewer_qoPY · 2024-10-31

**Soundness:** 4
**Presentation:** 3
**Contribution:** 3
**Rating:** 6
**Confidence:** 5

**Summary:**

In the paper, the MultiModal Few-Shot SegNet was proposed, which segments novel categories with minimal annotated support samples utilizing textual labels and the potentially available 2D image modality.  The main contributions are as follows: 1) To incorporate the textual modality of category names and the 2D image modality in few-shot 3D learning scenarios without additional annotation cost; 2) A new model, MM-FSS, was introduced, and information from different modalities was utilized. Experiments validated the value of the proposed method and the efficacy of the proposed method across different few-shot settings.

**Strengths:**

Based on the learning method, a large number of annotations are required for the identification and segmentation of new target categories in the point cloud, which is very time-consuming and laborious. The paper is of great significance to reduce the data annotation process through the combination of text and images.The MM-FSS model exhibits superior generalization capabilities, especially in few-shot settings, providing valuable insights for future research directions in the field.

**Weaknesses:**

1. The paper's description of the 2D image modality's integration into the multi-modal FS-PCS setup could be more detailed, providing clearer insights into how the 2D images are leveraged to enhance the model's performance without additional annotation costs.
2. The article does not offer a detailed analysis of the performance enhancements specifically attributable to the multi-modal information fusion during the modeling process, making it challenging to assess the individual impact of textual and 2D image modalities on few-shot segmentation performance.
3. The existing FS-PCS datasets are generally on a small scale, the practicability and effectiveness of the method in this paper need to be further verified in the future.

**Questions:**

1. In Line 110，ii should be iii.
2. Whether there is a comparison of the computational complexity and computational efficiency of different algorithms？
3. The significance of the important regions depicted in Figure 3 is unclear. The authors should provide additional details to enhance reader understanding.
4. The authors input multiple images into the VLM; how do they determine the viewpoints for the RGB images fed into the VLM?
5. The authors demonstrate a notable performance increase by introducing textual information into the query. I am curious if there is potential to also incorporate textual or other types of information into the support set to achieve further gains.
6. The selection of support appears somewhat random. If the authors could provide some theoretical guidance in the appendix regarding this selection process, I believe it would greatly enhance the significance of this work.

---

> ### Author Response · Authors · 2024-11-20
>
> Thank you for acknowledging the significance of our multimodal few-shot setup, the superior performance of our model, and the value of our work for advancing future research directions. Below, we address your questions:
>
> **W1: Insights for 2D modality**
>
> The 3D point cloud and 2D image modalities exhibit *distinct yet complementary* characteristics.
>
> 3D point clouds provide rich spatial geometry. However, they are inherently unstructured, lacking natural topology, and their sparsity—points distributed irregularly and concentrated on surfaces—limits the representation of fine-grained details.
>
> 2D images, in contrast, offer dense, structured representations encoding texture, color, and details on a pixel grid. However, they lack direct geometric cues about depth or the spatial structure of the scene.
>
> Despite their differences, the two modalities *share a natural correspondence*: a point in the 3D point cloud typically aligns with a pixel in a 2D image captured from the same perspective. This alignment allows combining the two modalities, laying the foundation for leveraging the strengths of the 2D modality to enhance 3D few-shot segmentation.
>
> Our approach leverages these insights to incorporate the 2D modality in an *implicit* manner during pretraining, *requiring no additional semantic labels*. Specifically, we use the visual encoder of VLMs to generate 2D visual features, which supervise 3D features from the IF head to simulate 2D features. Then, the learned 3D features serve as a *source of 2D information*, exploited to build multimodal understanding of novel classes during subsequent meta-learning and inference stages.
>
> Further pretraining details are provided in Appendix B. We will add more insights on the integration of 2D modality in the final version.
>
> **W2: Performance attribution of modalities**
>
> We analyzed the individual contributions of textual and image modalities to few-shot performance in Tab. 3d with explanations in Lines 499–503. These results present the *gains from each modality* and our model's effectiveness in *combining their complementary strengths* for a comprehensive understanding of novel classes. We will give it more emphasis in the final version.
>
> **W3: Dataset**
>
> The existing 3D datasets are relatively small in scale compared to 2D datasets. Following prior FS-PCS settings [2,3], we evaluated our method on standard benchmarks (ScanNet and S3DIS), which *align with existing methods*. We are happy to validate our approach on larger datasets as they become available in the future.
>
> **Q1: Typographical error**
>
> Thank you for noting this. We will correct “ii” to “iii” in the final version.
>
> **Q2: Complexity analysis**
>
> In the paper, Tab. 3g compares computational complexity with the state-of-the-art COSeg, with corresponding analysis in Lines 516–520.
>
> **Q3: Clarification on Figure 3**
>
> In Figure 3, the "important areas" refer to regions where predictions from COSeg and MM-FSS *differ significantly*. Colored circles highlight these regions to facilitate visual comparison. We will revise the caption to make the phrasing more explicit and avoid ambiguity.
>
> **Q4: Viewpoints for RGB images**
>
> During pretraining, we input images into the VLM to get the 2D features to supervise the 3D features. To establish the 2D-3D correspondences required for supervision, we use the camera-to-pixel intrinsic matrix and world-to-camera extrinsic matrix of the 2D images. These matrices are provided by the dataset since they are *typically known as part of the 3D point cloud capture process*. The detailed pretraining process is provided in Appendix B.
>
> **Q5: Textual information in the support set**
>
> In the TACC module, textual information has been used with the support set through the evaluation of $G_s$ (Eq. 10), which reflects the reliability of the semantic guidance $G_q$.
>
> Moreover, since our MSF module focuses on *modeling robust query-class correlations*, incorporating additional textual information in the support set *does not directly aid this process*. However, we agree that exploring effective designs to integrate different types of information into the support set for enhanced performance will be a valuable direction for future work.
>
> **Q6: Support set selection**
>
> Our meta-learning and inference follow the episodic paradigm [1], where each episode contains a support set and a query set. Following the episode construction in prior works [2,3], support sets are selected by randomly sampling a target class and choosing point clouds containing that class. Therefore, the support set is *designed to be selected randomly*. We will add a detailed illustration of this point to the final version.
>
> We hope our responses address your questions. Thank you again for your valuable feedback and tremendous efforts.
>
> [1] Matching networks for one shot learning. NeurIPS 2016.
>
> [2] Few-shot 3d point cloud semantic segmentation. CVPR 2021.
>
> [3] Rethinking Few-shot 3D Point Cloud Semantic Segmentation. CVPR 2024.

---

> > ### Comment · Reviewer_qoPY · 2024-12-03
> >
> > Thank you for the response. I have no further questions. I believe that the paper offers valuable insights to the community,

---

> ### Author Response · Authors · 2024-11-25
> **Kind Reminder: Discussion Phase Approaching Conclusion**
>
> Dear Reviewer qoPY,
>
> Thank you again for your positive review and valuable feedback on our paper.
>
> During the discussion period, we addressed your comments by:
>
> + Providing clearer insights into the utilization of the 2D modality and its integration.
> + Showing the individual performance contributions of each modality, as well as the computational complexity comparison (please refer to the detailed results in the mentioned locations of the paper, as noted in our response).
> + Clarifying questions regarding the datasets, figure caption, viewpoint usage, potential gains, and support set selection process.
> + Committing to correcting the typographical error and incorporating these discussions into the final version.
>
> As the discussion phase nears its conclusion, we kindly remind you to share any additional comments or questions you may have. We would greatly appreciate the opportunity to address any remaining concerns before the discussion period ends.
>
> Thank you for your time and consideration.
>
> Best regards,\
> Authors

---

### Official Review · Reviewer_k1NE · 2024-11-01

**Soundness:** 3
**Presentation:** 3
**Contribution:** 3
**Rating:** 8
**Confidence:** 3

**Summary:**

This work proposes a multimodal few-shot segnet (MM-FSS) to leverage multimodal information from textual labels and 2D images. Specifically, they propose multimodal correlation fusion (MCF) to fuse the multimodal correlations and enhance the fused features by multimodal semantic fusion (MSF) with textual information. Besides, the authors proposed test-time adaptive cross-modal calibration (TACC) to calibrate the predictions during test time. Experimental results on both S3DIS and ScanNet show the superior performance of the proposed method.

**Strengths:**

1. It makes sense to use complementary information from multimodal inputs for point cloud segmentation.
2. The ablation studies on ScanNet prove the effectiveness of the proposed models, including MCF, MSF, and TACC.
3. The features from 2D images are learned during the pre-training step. In this way, the proposed method does not rely on 2D inputs during inference.
4. The proposed MM-FSS achieves promising performance on two benchmark datasets compared to the state-of-the-art methods.

**Weaknesses:**

1. The idea of using multimodal information for 3D point cloud segmentation is not new. As noticed by the authors, such an idea has been explored for fully supervised or more challenging tasks like open-vocabulary segmentation. It is unclear why the prior few-shot methods only use unimodal information. It would be better to provide more discussions on the difficulties or challenges when using multimodal features for few-shot segmentation.

2. The claim about “cost-free” is ambiguous as “cost” may mean many things, such as training cost. I don't think it is necessary to claim the "cost-free".

**Questions:**

Please refer to the weakness.

---

> ### Author Response · Authors · 2024-11-20
>
> Thank you for acknowledging the reasonableness of our multimodal few-shot setup, the adequacy of our ablation studies, the independence of our models from 2D inputs during inference, and the promising performance of our approach. We provide our responses to your questions below:
>
> **W1: Challenges in few-shot segmentation**
>
> In fully supervised or open-vocabulary segmentation [1,2], the input is *a single point cloud* and the output is the *predicted semantics for that input*. The leverage of multimodality in these tasks focuses on enhancing feature representations of the individual input point cloud.
>
> In contrast, few-shot segmentation involves *multiple point cloud inputs* (support and query), where the goal is to segment novel classes in the query based on *knowledge derived from the support set*. The core of few-shot segmentation is to mine *meaningful connections between query and support point clouds* [3].
>
> Therefore, when incorporating multimodality in few-shot cases, how to effectively leverage multimodal features to *establish informative correlations* and *facilitate knowledge transfer from support to query* poses unique challenges.
>
> To this end, our proposed MCF and MSF modules effectively exploit visual and textual modalities to build *comprehensive multimodal connections*, and TACC uses cross-modal information to calibrate predictions, achieving more *robust knowledge transfer from support to query*.
>
> In the final version, we promise to include this discussion to provide additional context.
>
> **W2: The term "cost-free"**
>
> Thank you for pointing out the potential ambiguity of the term "cost-free". While our approach avoids the need for additional labelling effort, we acknowledge that "cost" may be interpreted more broadly (*e.g.*, computational or training cost). We will remove the term "cost-free" in the final version and revise the phrasing to avoid confusion.
>
> We hope the responses address your concerns. Thank you again for your valuable feedback and suggestions.
>
> [1] "Uniseg: A unified multi-modal lidar segmentation network and the openpcseg codebase." ICCV 2023.
>
> [2] "Openscene: 3D scene understanding with open vocabularies." CVPR 2023.
>
> [3] "Rethinking Few-shot 3D Point Cloud Semantic Segmentation." CVPR 2024.

---

> ### Author Response · Authors · 2024-11-25
> **Kind Reminder: Discussion Phase Approaching Conclusion**
>
> Dear Reviewer k1NE,
>
> Thank you again for your positive review and valuable feedback on our paper.
>
> During the discussion period, we addressed your comments regarding the challenges in few-shot segmentation by discussing the unique task formulation, its specific focuses, and the associated challenges. Additionally, we committed to following your suggestion to remove the term "cost-free" in the final version to avoid any ambiguity.
>
> As the discussion phase nears its conclusion, we kindly remind you to share any additional comments or questions you may have. We would greatly appreciate the opportunity to address any remaining concerns before the discussion period ends.
>
> Thank you for your time and consideration.
>
> Best regards,\
> Authors

---

> > ### Comment · Reviewer_k1NE · 2024-11-26
> >
> > Thanks for the detailed response. I have no more questions.

---

### Official Review · Reviewer_8jox · 2024-11-02

**Soundness:** 3
**Presentation:** 3
**Contribution:** 3
**Rating:** 8
**Confidence:** 3

**Summary:**

This paper studies the Few-Shot 3D Point Cloud Semantic Segmentation task by using multimodality information. Specifically, it uses 2D image modality to  extract additional intermodal features apart from the unimodal visual features. It also uses text embedding to refine the multimodal correlations generated by the proposed Multimodal Correlation Fusion module. Moreover, it presents the Test-time Adaptive Cross-modal Calibration technique to improve the generalization capability. The experimental results show that the proposed methods can achieve higher performance than previous works.

**Strengths:**

- The writing is good, making this paper easy to follow.

- The performance is good.

- The experiments are adequate.

- The introduction of the multimodality information is useful.

**Weaknesses:**

- How the design of the MSF module can achieve the improvement for the correlation. Please give more explanations.

**Questions:**

Please answer the question in weaknesses.

---

> ### Author Response · Authors · 2024-11-20
>
> Thank you for recognizing the quality of our writing, the adequacy of our experiments, the usefulness of introducing multimodal information, and the good performance of our proposed approach.
>
> Regarding your question, we provide the following detailed explanation about the MSF module:
>
> The MSF module enhances cross-modal visual correlations by incorporating textual semantic guidance, that is *supplementary to the visual guidance* derived from the support set. The detailed process is as follows:
>
> 1. MSF computes the affinities between intermodal query features and text embeddings, *producing the textual semantic guidance* $G_q$ (Eq. 5). It quantifies how well each query point relates to the target classes *in the text embedding space*, adding semantic context beyond visual correlations.
>
> 2. Recognizing that the relative importance of visual and textual modalities varies across points and classes, MSF *introduces point-category weights* $W_q$. These weights are determined based on information from both visual and textual modalities as in Eq. 6, dynamically weighting their contributions.
>
> 3. The weighted textual semantic guidance, $G_q \cdot W_q$, is then aggregated with the visual correlations to get *enhanced multimodal correlations* as in Eqs. 7 and 8.
>
> By leveraging supplementary textual guidance and dynamically balancing its contributions for each point and class, the MSF module results in *enhanced multimodal correlations*, effectively combining the strengths of textual and visual information to better determine the optimal class for query points.
>
> Furthermore, we provide *detailed visualizations of the effects of MSF* in Figure 5 of the supplementary material, with analysis in Lines 878–887. Figure 5 visualizes $G_q$ (5th column), $W_q$ (8th column), predictions without MSF (6th column), and predictions with MSF (7th column).
>
> We can observe the adaptive behaviour of MSF across different scenarios:
>
> - Case 1 (1st row): When target classes have *significant shape or appearance differences* between support and query samples, visual correlations alone struggle to establish meaningful connections (6th column). In such cases, $W_q$ assigns higher weights to regions emphasized by textual guidance (8th column), significantly improving the correlations and yielding more accurate predictions (7th column).
>
> - Case 2 (2nd row): When the target class is *visually similar between support and query samples*, $W_q$ is more evenly distributed across query points, maintaining a balanced contribution from both visual and textual modalities.
>
> The visualizations clearly demonstrate that the MSF module *adaptively combines the complementary strengths* of textual and visual modalities, effectively enhancing correlations and achieving more accurate segmentation.
>
> We will add further explanations in the final version to ensure clarity. We hope this technical illustration, supported by visualization analysis, clarifies the design and functionality of the MSF module. Thank you again for your insightful comments.

---

> ### Author Response · Authors · 2024-11-25
> **Kind Reminder: Discussion Phase Approaching Conclusion**
>
> Dear Reviewer 8jox,
>
> Thank you again for your positive review and valuable feedback on our paper.
>
> During the discussion period, we addressed your comments regarding the design of the MSF module by providing detailed technical explanations, outlining corresponding motivations, and analyzing the visualizations that illustrate the effects of MSF. Please refer to Figure 5 in the supplementary material for these visualizations. We also committed to adding further explanations in the final version to enhance clarity.
>
> As the discussion phase nears its conclusion, we kindly remind you to share any additional comments or questions you may have. We would greatly appreciate the opportunity to address any remaining concerns before the discussion period ends.
>
> Thank you for your time and consideration.
>
> Best regards,\
> Authors

---

### Official Review · Reviewer_YYsz · 2024-11-02

**Soundness:** 4
**Presentation:** 3
**Contribution:** 4
**Rating:** 8
**Confidence:** 5

**Summary:**

This paper investigates the few-shot 3D point cloud semantic segmentation task. It proposes a cost-free multimodal FS-PCS setup. Based on this setup, the authors introduce MM-FSS to utilize the multimodal information to help with few-shot learning. The model includes MCS to generate multimodal correlations and MSF to refine the correlations by textual guidance. To mitigate training bias, the authors also propose a TACC technique to adaptively calibrate predictions at test time. The experiments demonstrate its effectiveness.

**Strengths:**

+ The paper is well-written and easy to follow, with clear motivations for each proposed design.
+ It introduces the first multimodal few-shot 3D segmentation setting, which is cost-free and doesn’t increase labeling effort, consistent with standard few-shot tasks. The idea of leveraging available free modalities to help few-shot learning could provide valuable insights for the field.
+ The proposed model is novel and well-justified. The MCS and MSF modules use different modalities by exploiting cross-modal visual correlations and textual semantics, respectively. The TACC module is parameter-free and used at test time to calibrate predictions adaptively based on support samples.
+ The performance gains are substantial, and the experiments are thorough. Detailed ablation studies and visualizations are provided and effectively validate the proposed design choices and justify their motivations.

**Weaknesses:**

Certain paragraphs could be more concise. For example, the paragraph (lines 235-240) explains the two training steps. Then the paragraph (lines 259-266) writes about these two steps again. These two parts could be merged to improve conciseness.

**Questions:**

See weakness.

---

> ### Author Response · Authors · 2024-11-20
>
> Thank you for recognizing the strengths of our paper, including the clarity of writing, the value of the proposed multimodal few-shot 3D segmentation setup, the novelty and soundness of our model, and the significance of our performance gains validated by thorough experiments.
>
> Regarding your suggestion to combine our discussion on the two training steps (Lines 235–240 and 259–266):
>
> The content in Lines 235–240 serves as part of the Method Overview (Sec. 3.2) to provide readers with a *high-level understanding of our training pipeline*. This overview aims to set the context and prepare readers for the more detailed method explanations that follow.
>
> Subsequently, in Sec. 3.3, we delve into the extracted two feature sets and the training process for each. Specifically, the intermodal features are trained in the first step, while the unimodal features are trained during the second step. The paragraph in Lines 259–266 is included to clearly *articulate these differences in the context of visual feature extraction and training*.
>
> We understand your concern about potential redundancy, and we will follow your suggestion to make this part more concise in the final version. Thank you once again for your thoughtful review and valuable feedback.

---

> > ### Author Response · Authors · 2024-11-25
> > **Kind Reminder: Discussion Phase Approaching Conclusion**
> >
> > Dear Reviewer YYsz,
> >
> > Thank you again for your positive review and valuable feedback on our paper.
> >
> > During the discussion period, we addressed your comments regarding the two paragraphs, clarifying their distinct purposes and committing to revising this part in the final version to enhance conciseness.
> >
> > As the discussion phase nears its conclusion, we kindly remind you to share any additional comments or questions you may have. We would greatly appreciate the opportunity to address any remaining concerns before the discussion period ends.
> >
> > Thank you for your time and consideration.
> >
> > Best regards,\
> > Authors

---

> > > ### Comment · Reviewer_YYsz · 2024-12-02
> > >
> > > Thank you for the response. I have no further questions. I believe this paper offers valuable insights to the community, with clear contributions, and I therefore support its acceptance.

---

### Meta-Review · Area_Chair_eZUs · 2024-12-20

**Metareview:**

This paper introduces a new multi-modal Few-Shot 3D Point Cloud Semantic Segmentation (FS-PCS) task. The authors propose the MultiModal Few-Shot SegNet (MM-FSS) method, which effectively incorporates multi-modal information (2D images and text) in a few-shot learning scenario. This is achieved through the introduction of two key modules: the Multimodal Correlation Fusion (MCF) module and the Multimodal Semantic Fusion (MSF) module. To further improve generalization, MM-FSS also introduces a Test-time Adaptive Cross-modal Calibration (TACC) technique to mitigate training bias. The MM-FSS model demonstrates superior performance and generalization capabilities in few-shot settings.

Initial reviewer concerns focused on several areas, including:

- The conciseness of the writing (YYsz)
- A need for more discussion on the challenges of using multimodal features for few-shot segmentation (k1NE)
- Explanations regarding the effect of the MSF module in enhancing correlation (8jox)
- Insufficient implementation details, such as how 2D images are integrated into the multi-modal FS-PCS task, how viewpoints for RGB images are determined, and how support samples are selected (qoPY)
- The absence of an ablation study on the individual impact of textual and 2D image modalities (qoPY)
- The practical applicability and effectiveness of the method in large-scale FS-PCS datasets (qoPY)
- A comparison of computational complexity and efficiency (qoPY)
- The possibility of incorporating textual or other types of information into the support set (qoPY)

After the rebuttal, all reviewers acknowledged that their concerns had been addressed. As a result, the paper received positive ratings from all four reviewers. The AC concurs with the reviewers' assessments regarding the strengths of the paper, including the introduction of a practical multimodal few-shot 3D segmentation setting, the effectiveness of the proposed method, and the substantial performance improvements demonstrated.

Based on the positive feedback and the authors’ thorough response to the reviewers' concerns, the AC recommends acceptance of this paper.

**Additional Comments On Reviewer Discussion:**

Initial reviewer concerns focused on several areas, including:

- The conciseness of the writing (YYsz)
- A need for more discussion on the challenges of using multimodal features for few-shot segmentation (k1NE)
- Explanations regarding the effect of the MSF module in enhancing correlation (8jox)
- Insufficient implementation details, such as how 2D images are integrated into the multi-modal FS-PCS task, how viewpoints for RGB images are determined, and how support samples are selected (qoPY)
- The absence of an ablation study on the individual impact of textual and 2D image modalities (qoPY)
- The practical applicability and effectiveness of the method in large-scale FS-PCS datasets (qoPY)
- A comparison of computational complexity and efficiency (qoPY)
- The possibility of incorporating textual or other types of information into the support set (qoPY)

After the rebuttal, all reviewers acknowledged that their concerns had been addressed. As a result, the paper received positive ratings from all four reviewers. The AC concurs with the reviewers' assessments regarding the strengths of the paper, including the introduction of a practical multimodal few-shot 3D segmentation setting, the effectiveness of the proposed method, and the substantial performance improvements demonstrated.

---

### Decision · Program_Chairs · 2025-01-22

Accept (Spotlight)